# HIF-1α regulates IL-1β and IL-17 in sarcoidosis

**Jaya Talreja[1], Harvinder Talwar[1], Christian Bauerfeld[2], Lawrence I Grossman[3], Kezhong Zhang[3], Paul Tranchida[4], Lobelia Samavati[1]***

[1]Department of Internal Medicine, Division of Pulmonary, Critical Care and Sleep Medicine, Wayne State University School of Medicine and Detroit Medical Center, Detroit, United States; [2]Department of Pediatrics, Division of Critical Care, Wayne State University School of Medicine and Detroit Medical Center, Detroit, United States; [3]Center for Molecular Medicine and Genetics, Wayne State University School of Medicine, Detroit, United States; [4]Department of Pathology, Wayne State University School of Medicine and Detroit Medical Center, Detroit, United States

**Abstract** Sarcoidosis is a complex systemic granulomatous disease of unknown etiology characterized by the presence of activated macrophages and Th1/Th17 effector cells. Data mining of our RNA-Seq analysis of CD14[+]monocytes showed enrichment for metabolic and hypoxia inducible factor (HIF) pathways in sarcoidosis. Further investigation revealed that sarcoidosis macrophages and monocytes exhibit higher protein levels for HIF-α isoforms, HIF-1β, and their transcriptional co-activator p300 as well as glucose transporter 1 (Glut1). In situ hybridization of sarcoidosis granulomatous lung tissues showed abundance of HIF-1α in the center of granulomas. The abundance of HIF isoforms was mechanistically linked to elevated IL-1β and IL-17 since targeted down regulation of HIF-1α via short interfering RNA or a HIF-1α inhibitor decreased their production. Pharmacological intervention using chloroquine, a lysosomal inhibitor, decreased lysosomal associated protein 2 (LAMP2) and HIF-1α levels and modified cytokine production. These data suggest that increased activity of HIF-α isoforms regulate Th1/Th17 mediated inflammation in sarcoidosis.

DOI: https://doi.org/10.7554/eLife.44519.001

*For correspondence:
ay6003@wayne.edu

**Competing interests:** The authors declare that no competing interests exist.

## Introduction

Sarcoidosis is a systemic granulomatous disease of unknown etiology that is characterized by extensive local inflammation and granuloma formation in different organs with an increase in T-helper type 1 (Th1) mediated cytokine production (*Hunninghake et al., 1994*; *Iannuzzi et al., 2007*; *Miyara et al., 2006*; *Rastogi et al., 2011*). Pulmonary involvement in sarcoidosis is the leading cause of morbidity and mortality. In the lungs, the presence of activated macrophages and the expansion of oligoclonal T and B cells suggest sustained activation of inflammatory pathways in this disease (*Fazel et al., 1992*; *Iannuzzi et al., 2007*). Activated macrophages, monocytes, and T cells in sarcoidosis produce a plethora of cytokines including TNF-α, IL-1β, interferon (IFN) gamma, IL-17, and others (*Facco et al., 2011*; *Müller-Quernheim, 1998*; *Rastogi et al., 2011*; *Talreja et al., 2016*).

Previously, we have shown that sarcoidosis bronchoalveolar lavage (BAL) cells and alveolar macrophages (AMs), unlike those from healthy controls, exhibit high constitutively active p38 and lack dual specificity phosphatase (DUSP1 or MKP-1). The sustained p38 activation directly controls expression of several cytokines in sarcoidosis AMs and monocytes and the modulation of p38 regulates T cell responses (*Rastogi et al., 2011*; *Talreja et al., 2016*). Recently, we performed RNA-sequencing (RNA-seq) in sarcoidosis monocytes and identified altered gene expression profiles and cellular pathways (*Talreja et al., 2017*). These were: metabolic including glycolysis and lipolysis,

**eLife digest** Sarcoidosis is a rare disease that is characterized by the formation of small lumps known as granulomas inside the body. These lumps are made up of clusters of immune cells, and are commonly found in the skin, lung or eye. Other organs of the body can also be affected, and symptoms will vary depending on where in the body lumps form. There is currently no specific treatment for sarcoidosis, as the direct cause of the disease is unknown. The disease is often treated with drugs that suppress the immune system. However, this type of treatment can lead to significant side effects and patients will respond to these drugs in different ways.

Patients with sarcoidosis have a heightened immune response to microbes that can cause infections, and rather than providing protection, this heightened response causes damage and inflammation to the body's organs. Now, Talreja et al. have identified which genes and proteins control this inflammatory response in immune cells from the lungs and blood of sarcoidosis patients.

Immune cells in the lungs of sarcoidosis patients were found to have higher levels of hypoxia inducible factor (HIF) – a gene-regulating protein that controls the uptake and metabolism of oxygen in mammals. In addition, lung tissue affected with granulomas also expressed increased levels of a specific version of HIF known as HIF-1. Talreja et al. showed that the increased expression of HIF in the immune cells of sarcoidosis patients was mechanistically linked to the production of several molecules that promote inflammation. Inhibiting HIF-1 led to a decrease in the production of these inflammatory molecules, indicating that increased activity of HIF-1 causes inflammation in sarcoidosis patients.

It remains unclear what causes this abundance of HIF-1$\alpha$. It is possible that specific modifications of this factor prevent it from degrading, resulting in higher levels. By identifying a link between HIF-1 and inflammation, these findings open up potential new avenues of the treatment for sarcoidosis patients.

DOI: https://doi.org/10.7554/eLife.44519.002

phagocytosis, inflammation, oxidative phosphorylation, and HIF signaling pathways (*Talreja et al., 2017*). Among differentially expressed genes in sarcoidosis monocytes, we found a large number of genes containing hypoxia response elements (HREs) in their regulatory regions and, by pathway analysis, enrichment of hypoxia inducible factor signaling pathways. Furthermore, in an independent study applying $^1$H nuclear magnetic resonance (NMR)-based analysis, we identified metabolic and mitochondrial alterations in sarcoidosis (*Geamanu et al., 2016*). Based on these observations, we hypothesize that HIF-isoform expression plays an important role in the maintenance of inflammation (*Rastogi et al., 2011*; *Talreja et al., 2016*), metabolic imbalance, and mitochondrial dysfunction in sarcoidosis (*Geamanu et al., 2016*).

The oxygen-sensitive transcription factors HIF-1$\alpha$ and HIF-2$\alpha$ are key transcriptional regulators of hypoxia-associated genes to adapt to decreased availability of $O_2$ (*Semenza, 2011*; *Wang and Green, 2012*). In the presence of $O_2$, cytosolic HIF-$\alpha$ isoforms are hydroxylated by prolyl-hydroxylases (PHD) through an iron dependent mechanism, which prevents heterodimerization with HIF-1$\beta$ (ARNT) and consequent nuclear translocation as an active transcription factor (*Palazon et al., 2014*; *Semenza, 2003*; *Semenza, 2011*). HIF transcription factors alter the expression of various genes involved in metabolism, cell differentiation, proliferation, and angiogenesis in hypoxic tissues. Although the role of HIF-$\alpha$ isoforms in hypoxia and cancer is well studied, there is a knowledge gap regarding their role in regulating immune cells under normoxic conditions. The role of HIF-1$\alpha$ in sarcoidosis has not been studied. In the current study, we applied a combination of transcriptional and functional approaches to investigate the role of HIF-1$\alpha$ in mediating the inflammatory immune response in AMs, monocytes, and PBMCs of sarcoidosis patients as compared to healthy controls. Because sarcoidosis predominantly affects the lungs, we carried out the functional studies using AMs to determine the lung immune responses, while monocytes and PBMCs were used to assess peripheral immunity. Under normoxic conditions we found enhanced expression and activity of HIF-1$\alpha$ in sarcoidosis AMs and monocytes. Furthermore, HIF-1$\alpha$ expression was directly correlated with IL-1$\beta$ production in AMs and PBMCs. Down regulation of HIF-1$\alpha$ expression via short interfering

RNA (siRNA) decreased IL-1β in sarcoidosis AMs, while decreased HIF-1α expression in PBMCs decreased IL-1β and IL-17 in response to anti-CD3 challenge.

## Results

### RNA-seq data of sarcoidosis monocytes identifies enrichment of the HIF-1α signaling pathway

Patients (*Table 1* and Materials and methods) were ambulatory outpatients who were not hypoxic. Differentially expressed (DE) genes between sarcoidosis monocytes and healthy monocytes previously determined (*Talreja et al., 2017*) were subjected to pathway analysis. The pathway analysis showed impaction of metabolic pathways, including oxidative phosphorylation, purine and pyruvate metabolism in sarcoidosis. Because most of genes in these pathways showed the presence of hypoxia response elements (HREs), we further focused on interrogation of the HIF-pathway. *Figure 1A* shows the heat map of HIF signaling genes in monocytes. There are clear differences in the intensity and expression of genes related to the HIF pathway in monocytes of healthy controls and sarcoidosis subjects. Next, we compared the expression of selected genes related to HIF transcription factor activity. The transcription factor aryl hydrocarbon receptor nuclear translocator (ARNT, also known as HIF-1β) heterodimerizes with HIF-1α to form a transcriptional active complex (*Wolff et al., 2013*). The gene count between sarcoidosis and healthy control subjects demonstrate significantly higher ARNT gene expression in sarcoidosis monocytes (*Figure 1B*). Endothelial PAS domain protein 1 (EPAS1), also known as HIF-2α, is a hypoxia inducible transcription factor (*Hu et al., 2003*; *Thompson et al., 2014*). The EPAS1 gene count between sarcoidosis and healthy control subjects demonstrates significantly higher EPAS1 expression in sarcoidosis monocytes (*Figure 1C*). EP300 is a co-activator important for transcriptional activity of HIFs (*Palazon et al., 2014*). Similarly, we found

**Table 1.** Subject Demographics.

| Characteristic | Patients | Control subjects |
|---|---|---|
| Age, y | 27.7 ± 11.4 | 28 ± 8.4 |
| BMI | 29 ± 10.4 | 28 ± 3.6 |
| Gender, N (%) | | |
| Female | 35 (77) | 16(70) |
| Male | 10 (23) | 7 (30) |
| Race, N (%) | | |
| African American | 51 (100) | 15 (75) |
| Caucasian | 0 (0) | 5 (25) |
| CXR stage, N (%) | | |
| 0 | 0 (0) | NA |
| 1 | 5 (11) | NA |
| 2 | 30 (66) | NA |
| 3 | 10 (22) | NA |
| $O_2$ saturation at Room Air | 96–100 | 97–100 |
| Organ Involvements, N (%) | | |
| Neuro-ophtalmologic | 8 (17) | NA |
| Lung | 43 (95) | NA |
| Skin | 12 (26) | NA |
| Multiorgan | 26 (57) | NA |
| PPD | Negative | NA |

Definition of abbreviations: BMI = body mass index, CXR = chest X-ray, NA = not applicable, PPD = purified protein derivative

DOI: https://doi.org/10.7554/eLife.44519.005

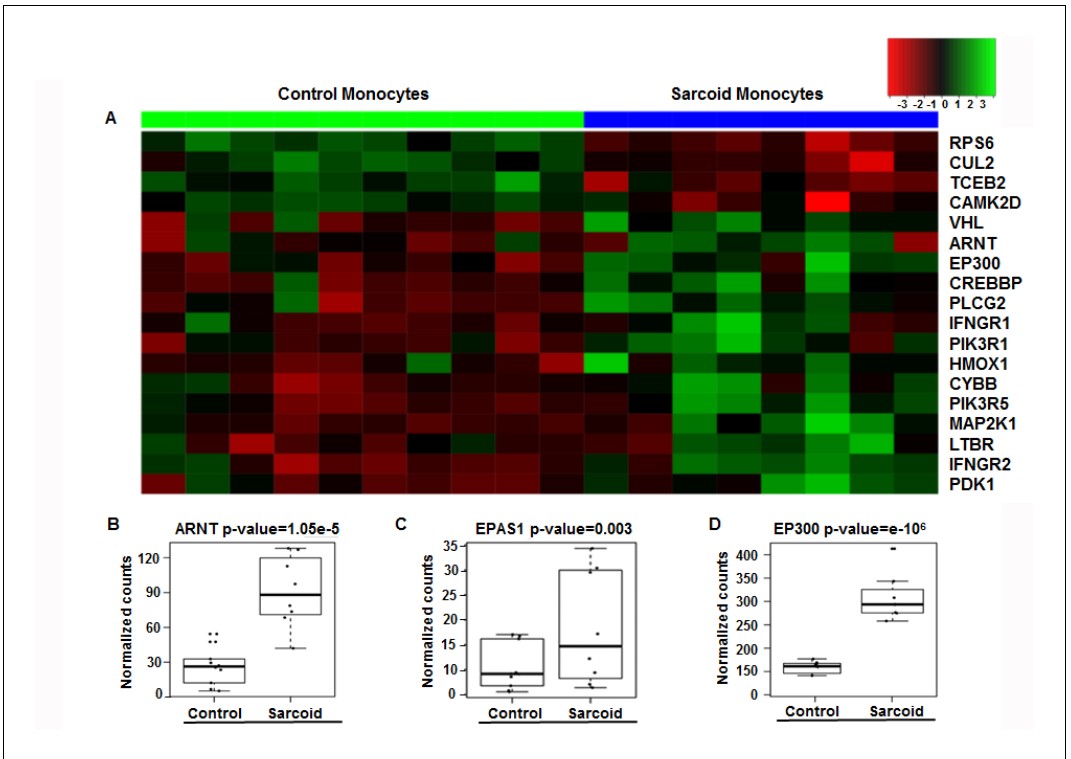

**Figure 1.** Enrichment of HIF-1α signaling pathways and related genes in sarcoidosis. Pathway analysis of DE genes between sarcoidosis versus healthy control monocytes was done using the iPathwayGuide tool. (**A**) Heatmap of genes involved in HIF-1α signaling between sarcoid and healthy control monocytes. Dendrograms according to means identifying genes levels in the heatmap show two distinct clusters. Green shading represents high expression and red shading represents low expression. (**B–D**) Data presented as box plots of gene counts corrected based on an FDR of 0.05. Boxplots for gene expression in monocytes are shown for ARNT (**B**), EPAS1 (**C**), and EP300 (**D**).

DOI: https://doi.org/10.7554/eLife.44519.003

The following source data is available for figure 1:

**Source data 1.** RNA-seq data of Sarcoid vs Healthy monocytes.

DOI: https://doi.org/10.7554/eLife.44519.004

higher p300 gene expression in sarcoidosis monocytes as compared to healthy controls (*Figure 1D*). However, there were no differences in HIF-1α gene transcripts between the two groups.

## Increased protein expression of HIF-α isoforms in sarcoidosis

Since HIF-1α is known to be predominantly regulated through modification of its protein stability (*Lee et al., 2004*; *Salceda and Caro, 1997*), we evaluated HIF-1α and HIF-2α protein abundance in AMs and monocytes of sarcoidosis patients, isolated as described in Materials and methods. AMs or monocytes were cultured ex vivo under normoxic conditions. Western analysis of cell lysates probed with antibody against HIF-1α showed increased HIF-1α protein expression in sarcoidosis AMs and monocytes (*Figure 2A and B*). Similar results were seen for HIF-2α protein expression (*Figure 2C and D*). Since HIFα heterodimerizes with ARNT (also known as HIF-1β), translocates to the nucleus, and recruits transcriptional coactivator p300 to transactivate target genes containing hypoxia-responsive elements (HREs) (*Semenza, 2003*; *Talwar et al., 2017a*; *Talwar et al., 2017b*), we also examined their protein expression. Sarcoidosis AMs also show a higher expression of ARNT (*Figure 2E and F*) and p300 (*Figure 2E and G*). Similarly, we evaluated the HIF-1α protein abundance in isolated monocytes from sarcoidosis subjects and healthy controls and found significantly higher HIF-1α expression (*Figure 2H and I*). However, in contrast to increased HIF-2α gene transcripts, we did not detect HIF-2α in either sarcoidosis or control monocytes at the protein level.

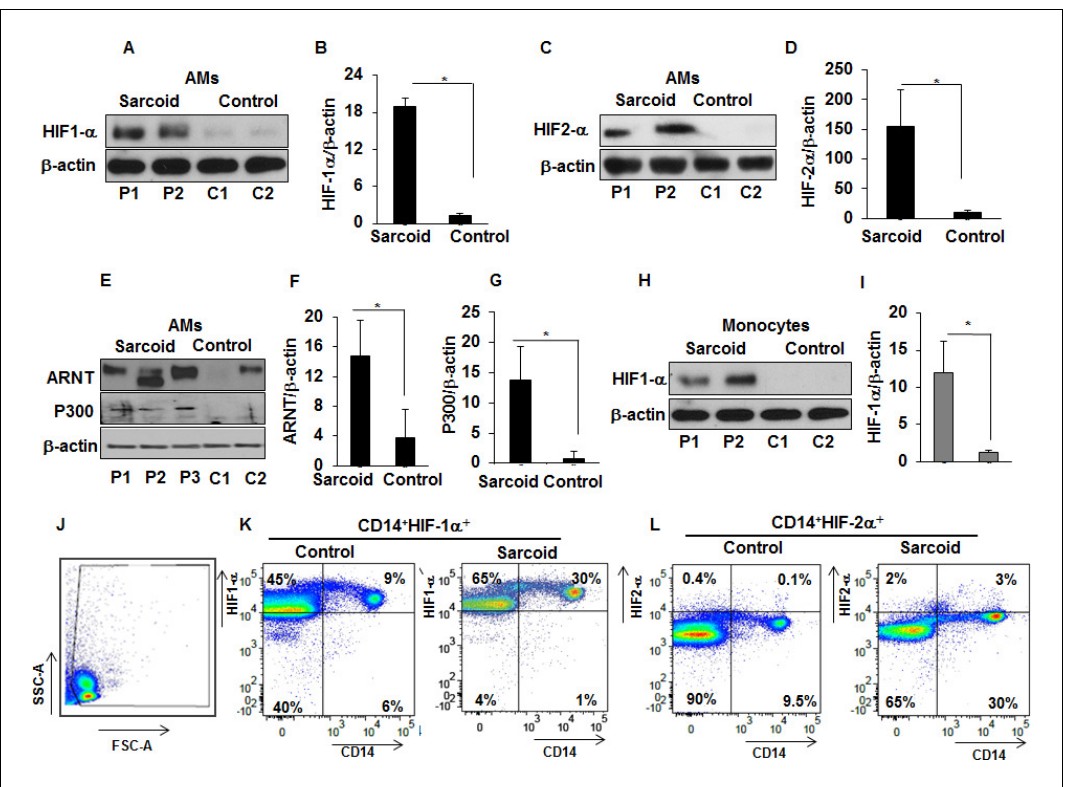

**Figure 2.** Increased expression of HIF-1α, HIF-2α, ARNT, and p300 in sarcoidosis. Whole cell extracts of AMs and monocytes were prepared and subjected to SDS-PAGE and western blot analysis using specific antibodies for HIF-1α, HIF-2α, ARNT, and p300. Equal loading was confirmed using antibodies against β-actin. Densitometry analysis is expressed as fold increase of the ratio of specific protein/β-actin. Sarcoid AMs exhibited higher HIF-1α expression (**A and B**) as compared to healthy controls. Sarcoid AMs expressed higher HIF-2α (**C and D**), ARNT (**E and F**) and p300 (**E and G**) as compared to healthy controls. Sarcoid monocytes exhibited higher HIF-1α expression (**H and I**) as compared to healthy controls. Representative blots for AMs and monocytes are shown out of a total of 10 patients and seven controls. Flow cytometry of PBMCs double stained for CD14+HIF-1α + and CD14+HIF-2α+ (**J and K**). PBMCs of healthy controls and sarcoid were stained with CD14-PerCPCy5.5, HIF-1α or CD14-PerCPCy5.5, HIF-2α primary antibodies followed by Alexa 488 secondary antibody and analyzed by flow cytometry using Flow-jo software. *Figure 2J* shows FSC-A/SSC-A gating. In healthy controls, 5–9% of PBMCs were CD14+HIF-1α+ whereas in sarcoidosis 20% to 35% of PBMCs were CD14+HIF-1α+ (**K**). HIF-2α expression was negligible in control PBMCs whereas 3% of sarcoid PBMCs were CD14+HIF-2α+ (**L**). Representative scatter plots from 4 patients and three controls are shown.

DOI: https://doi.org/10.7554/eLife.44519.006

Because the lack of detection could have been due to low protein abundance in monocytes or lower sensitivity of antibody epitope, we compared the HIF-1α and −2α expression by flow cytometry. *Figure 2J* shows FSC-A/SSC-A gating. FACS analysis of PBMCs double stained for CD14 and HIF-1α or HIF-2α shows that in healthy controls 5–9% of PBMCs are CD14+HIF-1α+, whereas in sarcoidosis 20% to 35% of PBMCs are CD14+ HIF-1α+. Analysis of CD14+ monocytes based on the expression of HIF-1α shows 25–60% HIF-1α+ CD14+ monocytes in controls, whereas in sarcoidosis HIF-1α+ CD14+ monocytes are 64–96% (**K**). Interestingly, in healthy controls 0–0.1% of PBMCs are CD14+ HIF-2α+, whereas in sarcoidosis 1–3% of PBMCs are CD14+ HIF-2α+. It shows that in healthy controls the percentage of HIF-2α+ CD14+ monocytes is negligible, whereas in sarcoidosis there is higher percentage of HIF-2α+ CD14+monocytes (5–9%) (*Figure 2L*). Thus, these results show that sarcoidosis AMs and peripheral monocytes exhibit increased expression of HIF isoforms compared to healthy controls. These data suggest a different protein expression profile of HIF-2α in lung macrophages versus peripheral monocytes with low abundance in monocytes versus AMs.

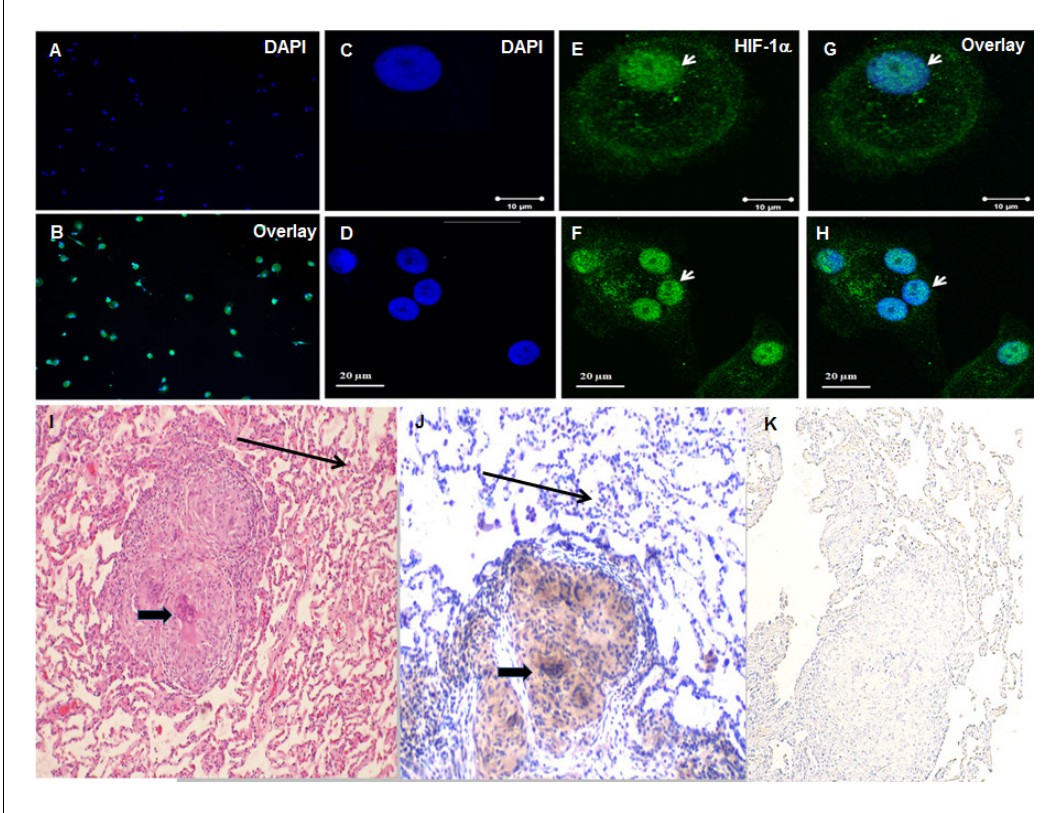

**Figure 3.** Increased HIF-1α expression in sarcoidosis AMs and granulomatous sarcoidosis lung tissue. Immunofluorescence staining of sarcoidosis AMs showing presence of HIF-1α in the cytoplasm and nuclei. AMs ($1 \times 10^5$) were allowed to adhere on chamber slides overnight. The cells were washed with PBST and fixed with 3.7% paraformaldehyde. Cells were permeabilized with 0.1% Triton X-100, blocked (10% FCS), and then incubated with anti-HIF-1α antibody overnight at 4°C. The secondary antibody was Alexa-fluor 488 - conjugated goat anti-rabbit antibody. Images were analyzed by immunofluorescent microscopy (AX10, Zeiss). Images show nuclei staining of AMs (**A**), overlay image shows nuclear and cytoplasmic co-localization of HIF-1α (**B**). Confocal laser scanning microscopy (CLSM-310, Zeiss) images show nuclei stained with DAPI (blue) in a single AM (**C**) and a multinucleated giant cell (**D**), nuclear and cytoplasmic accumulation of HIF-1α in green (**E and F**), overlay image shows nuclear co-localization of HIF-1α (**G and H**). The images are representative from two patients out of total of 5 patients. The photomicrographs represent in situ immunohistochemistry performed on lung tissues. H and E staining of tissue obtained from transbronchial biopsy (**I**) 100X, HIF-1α immunostaining (**J**), negative staining using isotype control antibody (**K**). The brown color represents an area of precipitate formed by a chromogenic substrate that is transformed by an enzymatic label conjugated to the antibody that has bound to the HIF-1α antigen. Note that the intensity of the staining is most pronounced in the histiocytic cells (i.e., AMs and the multinucleated giant cells, thick arrow), and is not identified in the surrounding alveoli (thin arrow). The immunohistochemistry images are representative from one patient out of total of 5 patients.

DOI: https://doi.org/10.7554/eLife.44519.007

The following figure supplement is available for figure 3:

**Figure supplement 1.** A 28- year-old woman underwent a liver biopsy for evaluation of increased transaminases.
DOI: https://doi.org/10.7554/eLife.44519.008

## Confocal microscopy of sarcoidosis AMs and immunohistochemistry of sarcoidosis tissues confirmed increased HIF-1α expression and its nuclear accumulation

To further confirm increased expression of HIF-1α protein in sarcoidosis and to determine whether HIF-1α accumulates in the nucleus, we immunostained AMs using specific an antibody against HIF-1α. Images were analyzed by immunofluorescent microscopy (AX10, Zeiss). We quantitated the percentage of cells showing HIF-1α expression (*Figure 3A and B*) in sarcoidosis. The staining is

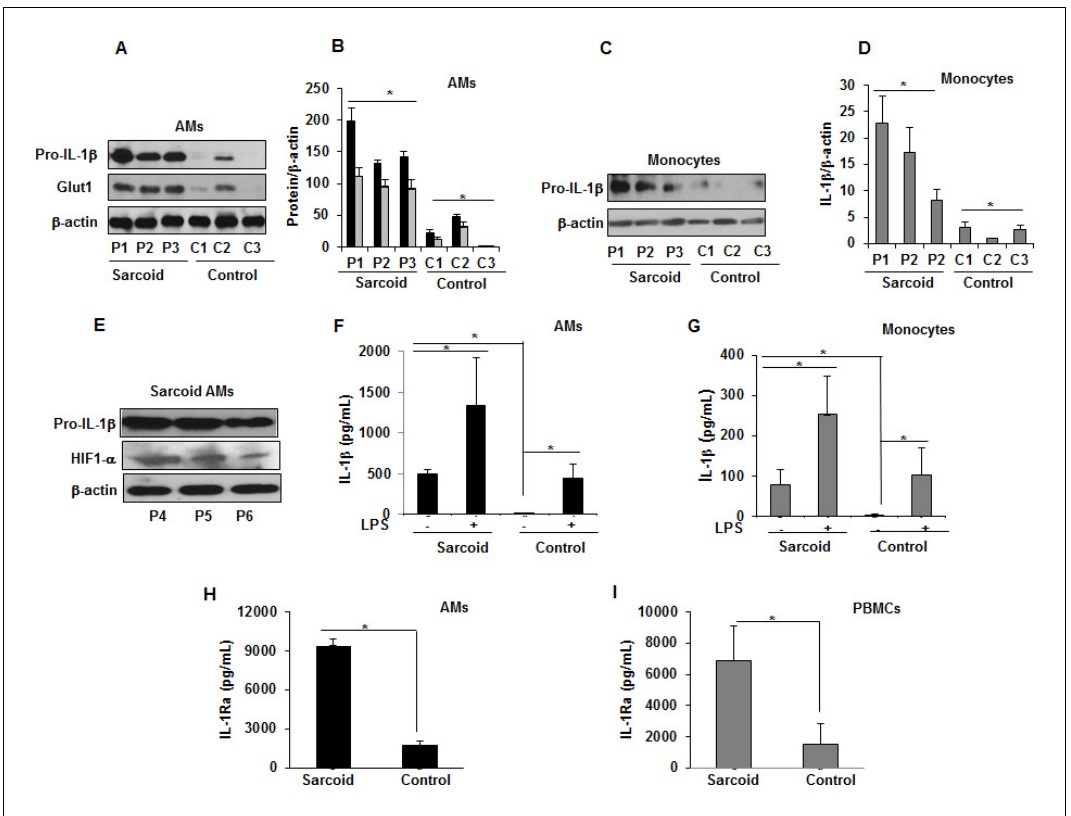

**Figure 4.** Increased Glut1, pro-IL-1β expression and IL-1β, IL-1Ra in sarcoidosis. AMs or monocytes from sarcoid subjects and controls were cultured overnight. Whole cell extracts were prepared, and culture supernatants were collected to measure IL-1β. Whole cell extracts were subjected to SDS-PAGE and western blot analysis using specific antibodies for Glut1, pro- IL-1β and HIF-1α. Equal loading was confirmed using β-actin antibody. Densitometry analysis is expressed as fold increase of the ratio of specific protein/β-actin. IL-1β was measured in culture supernatants via ELISA. Sarcoidosis AMs (n = 18) exhibited significantly higher expression of Glut1 and pro-IL-1β as compared to control subjects (n = 10) (**A and B**). The western blot and densitometric results (black bars for pro- IL-1β and grey bars for Glut1) are representative from three patients out of total of 18 patients and three controls out of total of 10 control subjects. Monocytes from sarcoid subjects also exhibited significantly higher pro-IL-1β as compared to controls (**C and D**). The western blot and densitometric results are representative from three patients out of total of 10 patients and 3 controls out of 10 control subjects. These data indicate that sarcoid AMs exhibit higher pro-IL-1β at baseline and this highly correlates with HIF-1α expression (**E**). Sarcoidosis AMs (**F**) and monocytes (**G**) produced significantly higher IL-1β cytokine at baseline and after LPS-stimulation as compared to healthy controls. Sarcoidosis AMs (**H**) and PBMCs (**I**) produced significantly higher IL-1Ra at baseline as compared to healthy controls. ELISA results represent mean ± SEM from 10 patients and 10 controls (4F and 4G), 10 patients and five controls (4H and 4I). *, p < 0.05 and was considered significant.
DOI: https://doi.org/10.7554/eLife.44519.009

The following source data is available for figure 4:

**Source data 1.** IL-1β and IL-1Ra production in sarcoid AMs,monocytes or PBMCs.
DOI: https://doi.org/10.7554/eLife.44519.010

representative of one out of the five patients. It shows that about 60–90% of AMs express HIF-1α. Images (*Figure 3C–H*) were analyzed by confocal laser scanning microscopy (CLSM-310, Zeiss). Confocal microscopy images show nuclei stained with DAPI (blue) in a single AM (C) and a multinucleated giant cell (D), nuclear and cytoplasmic accumulation of HIF-1α in green (E and F), overlay image shows nuclear co-localization of HIF-1α (G and H).We saw enhanced expression and accumulation of HIF-1α in the cytoplasm and nuclei of sarcoidosis AMs, both in a single AM (*Figure 3E*) and in a multinucleated giant cell (*Figure 3F*) that are known to be characteristic cells in sarcoidosis granuloma. HIF-1α is highly expressed and overlay images show that HIF-1α accumulates in nuclei (*Figure 3G and H*) as compared to cytoplasm. To further explore the expression seen in sarcoidosis AMs, we

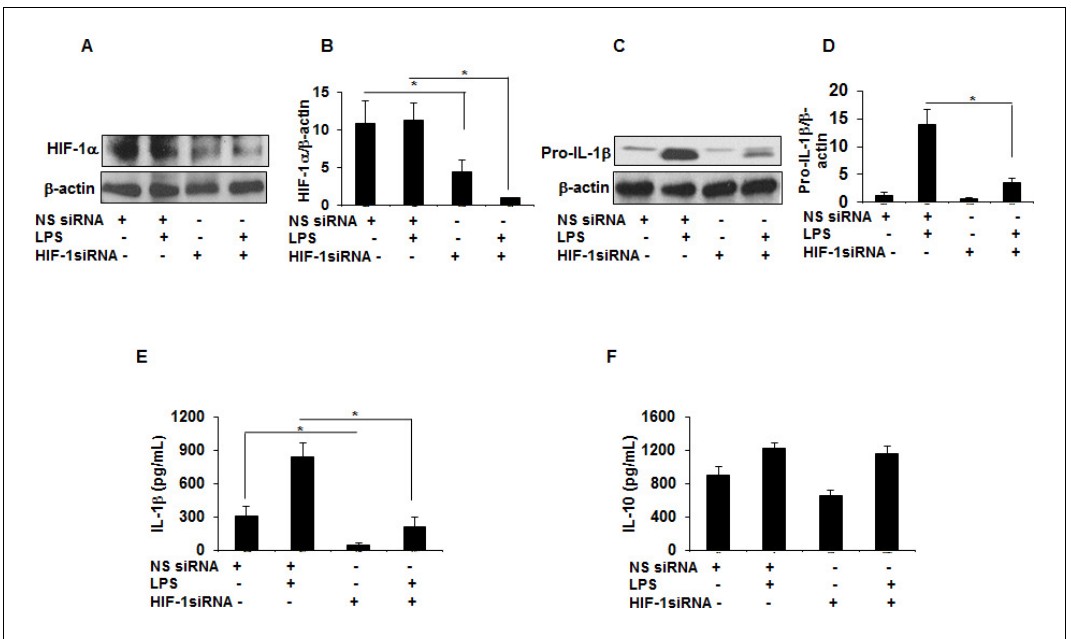

**Figure 5.** HIF-1α downregulation via siRNA decreases IL-1β production in sarcoid AMs. AMs from sarcoidosis subjects were transiently transfected with nonsense vector (NS siRNA, 200 pM) or targeted HIF-1α siRNA (200 pM, Thermofisher-Scientific). After 24 hr of transfection, cells were activated with LPS (100 ng/mL) for 3 hr. Whole cell lysates obtained after 3 hr of activation were subjected to immunoblotting to assess the HIF-1α and pro- IL-1β expression. Values were normalized to β-actin and are shown as relative expression to NS siRNA control. Conditioned media were collected after 24 hr and were assessed for different cytokines. HIF-1α siRNA significantly reduced both HIF-1α and pro-IL-1β protein in AMs (A–D). HIF-1α siRNA significantly inhibited IL-1β (E) but had no inhibitory effect on IL-10 (F) in AMs. Western blot data presented is a representative of four independent experiments. ELISA results represent mean ± SEM from four different experiments. *, p < 0.05 and was considered significant.

DOI: https://doi.org/10.7554/eLife.44519.011

The following source data is available for figure 5:

**Source data 1.** Effect of downregulation of HIF-1α via siRNA on IL-1β and IL-10 production in sarcoid AMs.
DOI: https://doi.org/10.7554/eLife.44519.012

---

assessed the presence of HIF-1α in lung biopsies of patients with sarcoidosis. Positive immunostaining was seen in multinucleated giant cells of granulomas as well as macrophages (*Figure 3I and J*, thick arrow), whereas fibroblasts and normal lungs lack HIF-1α expression. Negative staining was done by using isotype control antibody (*Figure 3K*). Similarly, we observed increased HIF-1α immunostaining signal in sarcoidosis liver and skin tissue samples. These results further confirmed that HIF-1α accumulates in sarcoidosis granulomatous tissues.

## Increased Glut1, pro-IL-1β levels and IL-1β, IL-1Ra production in sarcoid AMs and monocytes

HIF-1α is a critical transcription factor regulating metabolic reprogramming during inflammation, in part through upregulation of the *SLC2A1* gene encoding glucose transporter (Glut)1 (*Chen et al., 2001*). HIF-1α and Glut1 upregulation contribute to production of several pro-inflammatory cytokines including IL-1β (*Talwar et al., 2017a*; *Talwar et al., 2017b*; *Tannahill et al., 2013*). Therefore, we evaluated the expression of Glut1 and pro-IL-1β at baseline in AMs and monocytes from sarcoidosis and control subjects. Sarcoidosis AMs exhibited a variable amount of Glut1 and pro-IL-1β (18/18 patients) but only 1 out of 10 healthy controls showed expression (*Figure 4A and B*). We found similar results for pro-IL-1β in monocytes (*Figure 4C and D*). Furthermore, increased pro-IL-1β expression directly correlated with Glut1 and HIF-1α expression in sarcoidosis AMs (*Figure 4E*). To determine whether increased pro-IL-1β expression in sarcoidosis leads to released IL-1β, we

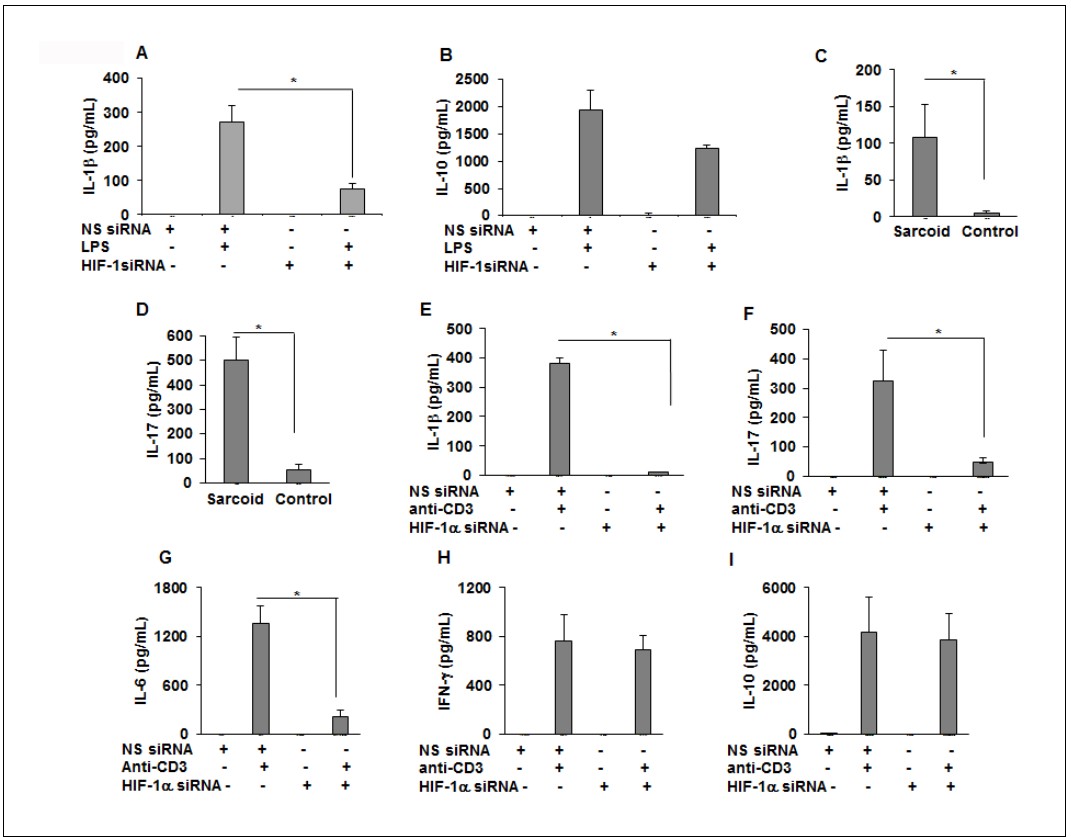

**Figure 6.** Downregulation of HIF-1α reduces the production of IL-1β, IL-17, and IL-6 in sarcoid PBMCs. PBMCs were transiently transfected with nonsense vector (NS siRNA, 200 pM) or targeted HIF-1α siRNA (200 pM, Thermofisher-Scientific). After 24 hr of transfection, cells were activated with either LPS (100 ng/mL) or anti-CD3 (1 μg/mL) in the presence of rhIL-2 (10 ng/mL). Conditioned media were collected after 24 hr (stimulated with LPS) or after 72 hr (stimulated with anti-CD3) and were assessed for cytokines via ELISA. HIF-1α siRNA significantly inhibited IL-1β (A) but had no inhibitory effect on IL-10 (B). The conditioned media of anti-CD3 stimulated sarcoidosis PBMCs (n = 11) or healthy control PBMCs (n = 10) show that sarcoidosis PBMCs produced significantly higher IL-1β (C) and IL-17 (D) as compared to healthy control PBMCs. HIF-1α siRNA significantly inhibited IL-1β (E), IL-17 (F) and IL-6 (G). HIF-1α siRNA did not inhibit IFN-γ (H), or IL-10 (I). ELISA results obtained from siRNA experiments represent mean ± SEM of four different experiments. *, p < 0.05 and was considered significant.
DOI: https://doi.org/10.7554/eLife.44519.013

The following source data is available for figure 6:

**Source data 1.** Effect of downregulation of HIF-1α via siRNA on IL-1β , IL-10, IL-17,IL-6 and IFN-γ production in sarcoid PBMCs.
DOI: https://doi.org/10.7554/eLife.44519.014

---

measured secreted IL-1β in the conditioned media of AMs and monocytes cultured in the absence or presence of LPS via ELISA. The results showed that unstimulated and LPS-stimulated cultured sarcoidosis AMs and monocytes secrete higher IL-1β as compared to healthy controls (*Figure 4F and G*). These data suggest that increased expression of HIF-1α leads to increased IL-1β production in sarcoidosis patients. The interleukin one receptor antagonist (IL-1Ra) is mainly secreted by monocytes, macrophages, and neutrophils. IL-1Ra (IL-1RII) competitively binds to IL-1β and forms a non-signaling complex IL-1Ra to the surface receptors for IL-1β and inhibits the effect of IL-1β on cells (*Arend, 2000*; *Janson et al., 1991*). Since the sarcoidosis AMs produced significantly high levels of IL-1β, we assessed the conditioned media for the secreted IL-1Ra. *Figure 4H* shows that sarcoidosis AMs produced significantly high levels of IL-1Ra as compared to control AMs. Similarly, sarcoidosis PBMCs (*Figure 4I*) produced high levels of IL-1Ra as compared to control PBMCs.

## Targeted downregulation of HIF-1α decreases IL-1β production in sarcoidosis AMs

IL-1β is regulated at the transcriptional level through expression of several transcription factors including Signal Transducer and Activator of Transcription (STAT) 3, HIF-1α, and others (*Samavati et al., 2009*; *Talwar et al., 2017b*). To determine the relative contribution of increased HIF-1α in IL-1β production in sarcoidosis AMs, we transiently transfected sarcoidosis AMs with either non-targeted siRNA or HIF-1α targeted siRNA. After 24 hr of transfection, cells were treated with LPS (100 ng/mL). Targeted downregulation of HIF-1α via siRNA of sarcoidosis AMs led to a significant reduction (about 50%) in HIF-1α (*Figure 5A and B*) and pro-IL-1β (*Figure 5C and D*) protein expression. To determine the specificity of targeted downregulation of HIF-1α on other cytokines, we assessed the conditioned medium for IL-1β and IL-10 production and found significantly decreased IL-1β production (*Figure 5E*). However, HIF-1α inhibition did not inhibit IL-10 production (*Figure 5F*).

## Downregulation of HIF-1α modulates cytokine profiles in sarcoidosis PBMCs in response to LPS and anti-CD3

Similar to AMs, the targeted down regulation of HIF-1α in sarcoidosis PBMCs resulted in decreased production of IL-1β in response to LPS (*Figure 6A*); the effect of HIF-1α inhibition was specific for IL-1β since there was no significant effect on IL-10 production (*Figure 6B*). These results clearly show that HIF-1α expression regulates IL-1β production in sarcoidosis AMs and PBMCs.

Recent work has shown that the HIF transcription factors are key elements in the control of immune cell metabolism and function in macrophages, B-cells, and T-cells (*Palazon et al., 2014*; *Wang and Green, 2012*). T helper 17 cells (Th17) represent a lineage of effector T cells critical in host defense and autoimmunity. It is has been shown that Th1 and Th17 cells contribute to sarcoidosis pathology (*Ramstein et al., 2016*). Based on this, we hypothesize that the HIF-1α inhibition may also modulate IL-1β and IL-17 production in response to anti-CD3 challenge. First, we assessed the effect of anti-CD3 activation on the production of IL-1β and IL-17 in healthy controls and sarcoidosis PBMCs. PBMCs were treated with anti-CD3 for 24 hr and the conditioned media were assessed for IL-1β and IL-17 production. Sarcoidosis PBMCS were seen to produce significantly higher levels of IL-1β (*Figure 6C*) and IL-17 (*Figure 6D*). To investigate the contribution of HIF-1α in Th1/Th17 cytokine production, we investigated the effect of targeted downregulation of HIF-1α in PBMCs in response to anti-CD3 challenge on the production of various inflammatory cytokines. Inhibition of HIF-1α by siRNA significantly decreased the production of anti-CD3 induced IL-1β (*Figure 6E*), IL-17 (*Figure 6F*), and IL-6 (*Figure 6G*). However, inhibition of HIF-1α did not decrease IFN-γ (*Figure 6H*) and IL-10 (*Figure 6I*) production. These results suggest that HIF-1α specifically regulates IL-1β and IL-17 in sarcoidosis.

## Pharmacological HIF-1α inhibition decreases the percentage of activated T-cells and cytokines in sarcoidosis PBMCs in response to anti-CD3

To confirm our results, we used echinomycin, a small molecule inhibitor of HIF-1α that has been shown to inhibit HIF-1α DNA binding activity (*Tang and Yu, 2013*; *Vlaminck et al., 2007*). We evaluated the effect of echinomycin HIF-1α inhibition on anti-CD3-induced IL-1β and IL-17 production and T cell activation in sarcoid PBMCs. To do so, cultured sarcoidosis PBMCs were pre-treated with echinomycin in vitro, then activated with anti-CD3 in the presence of rIL-2, followed by determination of activated CD4$^+$CD25$^+$ T-cells by flow cytometry and measurement of cytokines by ELISA. Our results showed that PBMCs of patients with sarcoidosis (n = 23) exhibit higher expression for activated CD4$^+$CD25$^+$T cells (mean ± SEM, 11.08 ± 5.32% as compared to healthy (n = 7) controls (mean ± SEM, 5.16 ± 2.71%, p < 0.05). *Figure 7A* shows that PBMCs of a patient with sarcoidosis exhibited higher expression for activated CD4$^+$CD25$^+$T cells (10%), further increasing to 50% in response to anti-CD3 stimulation (*Figure 7B*). Pre-treatment of PBMCs with echinomycin decreased the number of activated T cells (3%) at base line (*Figure 7C*) and in response to anti-CD3 stimulation to 15% (*Figure 7D*). Furthermore, pretreatment with echinomycin significantly decreased both baseline and anti-CD3 induced IL-1β production (*Figure 7E*). Similarly, pretreatment with echinomycin

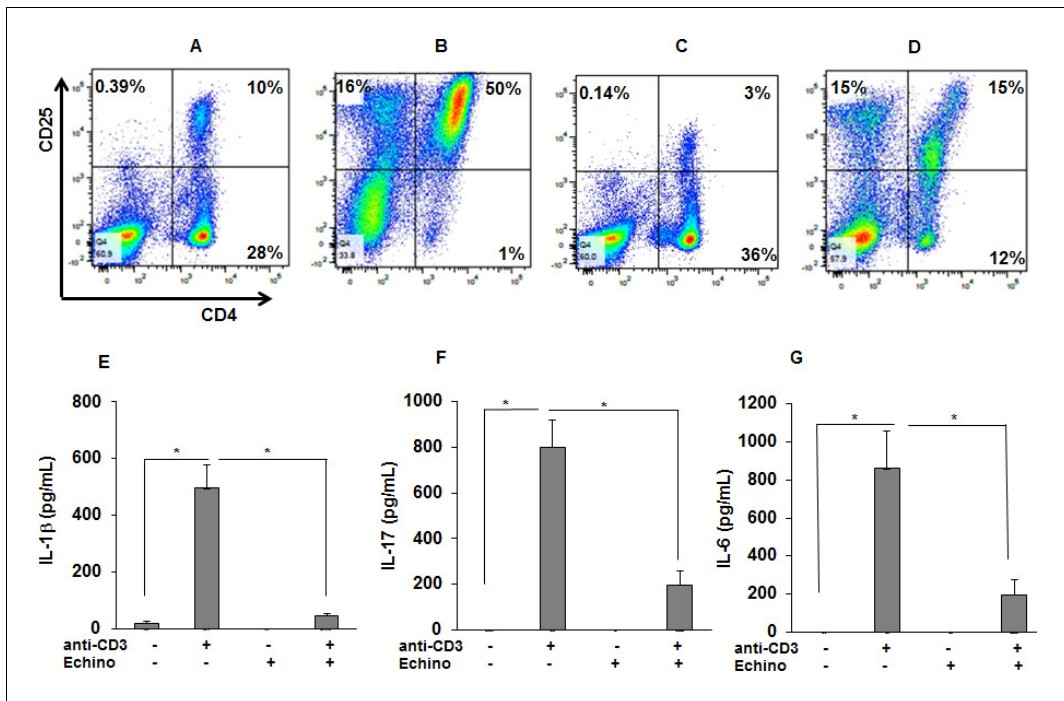

**Figure 7.** HIF-1α inhibition reduces the percentage of activated CD4 +CD25+cells in anti-CD3 stimulated sarcoid PBMCs and the production of IL-1β, IL-17, and IFN-γ. PBMCs of sarcoid subjects were pretreated with echinomycin (HIF-1α inhibitor, 10 nM) for 30 min and were stimulated with anti-CD3 (1 μg/mL) in the presence of rhIL-2 (10 ng/mL) for 72 hr. Cells were harvested after 72 hr of culture and immunostained with fluorescein conjugated antibodies CD4 and CD25 and analyzed by flow cytometry using Flow-jo software. (A–D) Representative scatter plots show FACS analysis of CD4 and CD25 expression of sarcoidosis PBMCs. The percentage of CD4 and CD25 double positive, representing activated T-cells, were 10% in untreated PBMCs (A). In sarcoidosis PBMCs stimulated with anti-CD3 the percentage of CD4 and CD25 double positive T-cells increased to 50% (B). Sarcoidosis PBMCs cultured in the presence of echinomycin for 72 hr. The percentage of CD4 and CD25 double positive cells decreased from 10% to 3% (C). Sarcoidosis PBMCs were stimulated with anti-CD3 in the presence of echinomycin. The percentage of activated T-cells decreased from 50% after anti-CD3 challenge to 15% in the presence of echinomycin (D). Data presented is a representative plot of 5 independent experiments. The conditioned medium was assessed for IL-1β, IL-17 and IFN-γ using ELISA. Echinomycin significantly inhibited anti-CD3-induced IL-1β (E), IL-17 (F) and IFN-γ (G). Data represent mean ± SEM from six different experiments. *, p < 0.05 and was considered significant.

DOI: https://doi.org/10.7554/eLife.44519.015

The following source data is available for figure 7:

**Source data 1.** HIF-1α inhibition reduces the production of IL-1β, IL-17, and IFN-γ in anti-CD3 stimulated sarcoid PBMCs.
DOI: https://doi.org/10.7554/eLife.44519.016

significantly decreased anti-CD3 induced IL-17 (*Figure 7F*) and IL-6 (*Figure 7G*) production in sarcoidosis PBMCs.

## Chloroquine modifies LAMP2, HIF-1α protein expression and inhibits IL-1 β and IL-17 production in sarcoidosis

Chloroquine (CHQ) is an anti-malarial drug and remains an integral treatment for systemic inflammatory diseases such as systemic lupus erythematosus and sarcoidosis (*Lee et al., 2011*; *Morse et al., 1961*). CHQ inhibits lysosomal degradation/autophagy either by altering lysosomal acidification or inhibiting the levels of lysosomal associated proteins (LAMP) (*He et al., 2011*; *Ma et al., 2012*; *Rubinsztein et al., 2007*). We hypothesized that CHQ modulates LAMP2, HIF-1α, and HIF-2α levels and cytokine production in sarcoidosis AMs and PBMCs. To examine this hypothesis, isolated AMs were pre-treated with CHQ and then activated with LPS. Interestingly, CHQ decreased LAMP2 levels

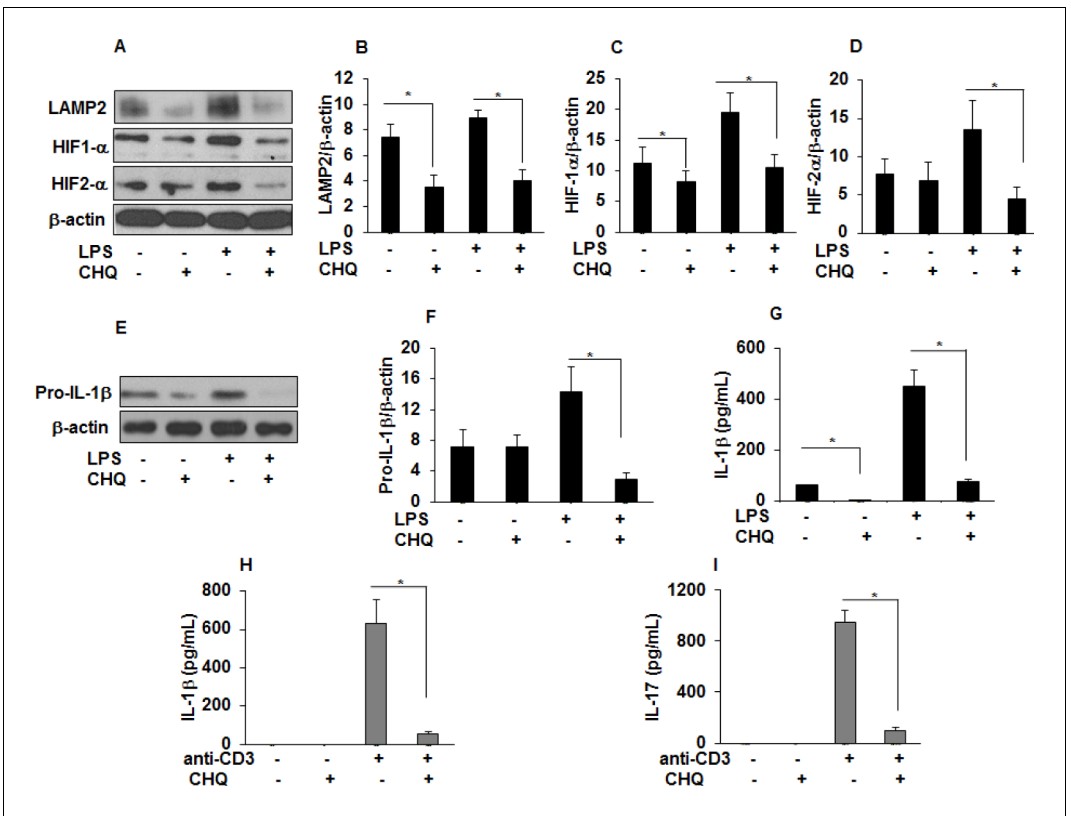

**Figure 8.** Chloroquine (CHQ) decreases LAMP2, HIF-α, IL-1 β, and IL-17 production in sarcoidosis. Sarcoidosis AMs were pretreated with CHQ (100 μM) for 30 minutes and activated with LPS (100 ng/mL) for 3 hours. Whole cell lysates obtained after 3 hours of activation were subjected to immunoblotting to assess the LAMP2, HIF-1α, HIF-2α, and pro-IL-1β expression. Values were normalized to β-actin and are shown as relative expression to untreated cells. Densitometry analysis is expressed as fold increase of the ratio of specific protein/ β-actin. Culture supernatants were assessed for IL-1β via ELISA. CHQ significantly inhibited LAMP2 (50%) at baseline and both HIF-1α (50%) and HIF-2α (50%) protein expression in response to LPS challenge (**A-D**). CHQ significantly inhibited pro-IL-1β (70%) protein expression in response to LPS treatment (**E and F**) and significantly inhibited IL-1β production both at baseline and in response to LPS (**G**). Sarcoid PBMCs were pretreated with CHQ (100 μM) for 30 min and activated with anti-CD3 (1 μg/mL) in the presence of rhIL-2 (10 ng/mL). Conditioned media were collected after 72 hours and were assessed for IL-1 β and IL-17 via ELISA. CHQ significantly inhibited anti-CD3 induced IL-1β (**H**) and IL-17 (**I**) production. Western blot data presented is representative of five independent experiments. ELISA results represent mean ± SEM from five different experiments. *, p < 0.05 and was considered significant.
DOI: https://doi.org/10.7554/eLife.44519.017

The following source data is available for figure 8:

**Source data 1.** Chloroquine (CHQ) decreases IL-1 β and IL-17 production in sarcoidosis..
DOI: https://doi.org/10.7554/eLife.44519.018

and both HIF-1α (by approximately 50%) and HIF-2α protein expression (by approximately 65%) in sarcoidosis AMs after LPS stimulation (*Figure 8A–D*). Furthermore, CHQ significantly decreased (70%) the expression of pro-IL-1β (*Figure 8D and E*). Similarly, measurement of released IL-1β in conditioned medium was significantly decreased both at baseline and in response to LPS stimulation (*Figure 8F*). To assess the effect of CHQ on IL-1β and IL-17 production by sarcoidosis PBMCs, cultured PBMCs were pre-treated with CHQ in vitro and then activated with anti-CD3. CHQ significantly decreased anti-CD3 induced IL-1β (*Figure 8G*) and IL-17 (*Figure 8H*) production in sarcoidosis PBMCs (p < 0.05).

## Discussion

Sarcoidosis is a chronic granulomatous disease with aberrant immune response to undefined environmental or infectious triggers (*Iannuzzi et al., 2007*). How specific antigens lead to a sustained granulomatous inflammation in sarcoidosis is largely unknown. Our novel RNA-seq data showed aberrant metabolic pathways and enrichment of DE genes for HIF pathways in monocytes of sarcoidosis patients (*Talreja et al., 2017*), confirming our previous metabolomics data showing aberrant metabolic pathways including increased glycolysis and malfunctional tricarboxylic acid (TCA) cycle in sarcoidosis (*Geamanu et al., 2016*; *Talreja et al., 2017*). In the current study, we investigated the role of HIF-isoforms in sarcoid alveolar macrophages and blood monocytes as well as PBMCs. Alveolar macrophages and monocytes have a central role in the maintenance of immunological homeostasis in response to pathogens providing an important host-defense (*Aberdein et al., 2013*). In sarcoidosis, both cell types are in an activated state and produce spontaneous ex vivo cytokines and chemokines including, IL-1β, TNF-α, IL-6, IL-18, and others (*Gracie et al., 2003*; *Müller-Quernheim, 1998*; *Rastogi et al., 2011*; *Rolfe et al., 1993*). Our current study confirms our previous findings that IL-1β plays an important role in sarcoidosis (*Rastogi et al., 2011*; *Talreja et al., 2016*). In addition, we find increased IL-1Ra in sarcoidosis AMs and PBMCs, suggesting activation of the IL-1 pathway. IL-1Ra is a member of the IL-1 family, whose production is stimulated by many substances including cytokines and bacterial or viral components; it has been suggested to act as a decoy receptor and is a natural inhibitor for the biologically active IL-1β (*Lang et al., 1998*); (*Arend, 2000*; *Santarlasci et al., 2013*). In several inflammatory diseases, including lupus and Crohn's disease (CD), elevated IL-1β production is associated with IL-1Ra (*Cominelli and Pizarro, 1996*). Our data are in line with previous studies showing increased IL-1Ra in sarcoidosis (*Mikuniya et al., 2000*; *Rolfe et al., 1993*). Further studies need to delineate the clinical role of IL-1Ra in sarcoidosis.

Here, we show that sarcoidosis AMs and monocytes in normoxic ex vivo culture conditions and without any stimulation exhibit constitutively active HIF-1α and HIF-1β (ARNT) along with its coactivator, p300. Furthermore, in situ HIF-1α immune staining of sarcoidosis lung biopsies demonstrated HIF-1α abundance in the center of granulomatous tissue and in multinucleated giant cells. We found that a higher percentage of CD14[+] monocytes express HIF-1α and HIF-2α in sarcoidosis subjects as compared to controls. Our data show that the increased HIF-1α expression is coupled to increased Glut1 protein levels, and enhanced IL-1β, IL-6 and IL-17 production. Downregulation of HIF-1α via siRNA or chemical inhibitors in sarcoidosis PBMCs leads to a decrease in IL-6 and IL-17 production at baseline and in response to anti-CD3 stimulation. In sarcoid subjects HIF-2α was predominantly expressed in the lung macrophage population whereas sarcoidosis monocytes showed lower levels of HIF-2α. HIF-2α downregulation had no significant effect on IL-1β and IL-17 production in sarcoidosis (data not shown). We speculate that HIF-2α regulates other macrophage functions such as phagocytosis and cell metabolism. Classically, sarcoidosis granulomas feature activated antigen presenting cells initiating adaptive immune responses with an increase in activated CD4[+]T-cells and Th1 mediated cytokines. Recently, it has been shown that Th17[+]/CD4[+]T cells are increased in sarcoidosis granulomatous tissue and peripheral blood (*Facco et al., 2011*; *Ostadkarampour et al., 2014*; *Ramstein et al., 2016*). Recent studies indicated that IL-1β plays a critical role in regulation of Th1/Th17 cells in response to commensal microbes (*Duhen and Campbell, 2014*). IL-1β promotes Th17 differentiation from naive CD4[+] T cells by enhancing IL-1 receptor expression (*Lee et al., 2010*). Furthermore, IL-1 synergizes with IL-6 to regulate Th17 differentiation and effector Th17 cell function through regulation of transcription factors, including IRF4 and RORγt (*Chung et al., 2009*). Thus, in sarcoidosis increased IL-1β and IL-6 explains Th17 differentiation. Previously, our group and other showed increased IL-6 production in AMs and PBMCs of sarcoidosis subjects at baseline and in response to TLR or NLR ligands (*Rastogi et al., 2011*; *Talreja et al., 2016*). Levels of IL-6 may be important in progression of fibrotic lung changes in sarcoidosis (*Le et al., 2014*). Our data indicate that downregulation of HIF-1α via siRNA or chemical inhibitor reduces IL-6 production by sarcoid PBMCs.

HIF-1α and HIF-2α are two critical transcription factors that regulate an array of genes involved in inflammation, angiogenesis, metabolic reprogramming, mitochondrial function, T-cell differentiation and Th17 development (*Cummins et al., 2016*; *Dang et al., 2011*; *Nizet and Johnson, 2009*; *Palazon et al., 2014*; *Phan and Goldrath, 2015*). Upregulation of HIF isoform plays a critical role in providing metabolic reprogramming in myeloid cells that is required to develop trained immunity

for a robust immune response (*Cheng et al., 2014*). It has been shown that mice with a myeloid cell-specific defect in HIF-1α were unable to mount a trained immune response against bacterial sepsis (*Cheng et al., 2014*; *Netea et al., 2016*). Trained immunity is associated with profound metabolic reprogramming in macrophages (*Yao et al., 2018*), dendritic cells, and natural killer cells (*Netea et al., 2016*). New mounting evidence indicates that metabolic reprogramming, including upregulation of glycolysis and depression of the TCA cycle, is a required metabolic switch for the development of innate memory, which in turn leads to upregulation of inflammatory cytokines including IL-1β and IL-17. Similar to cancer metabolism, during inflammation aerobic glycolysis (Warburg effect) plays an important role in the maintenance of cellular energy supply (*Koppenol et al., 2011*; *Warburg, 1956*). Sarcoidosis AMs and monocytes exhibit a phenotype resembling the Warburg effect or trained immunity exhibiting an abundance of HIF isoforms, higher expression for Glut1, and higher production of IL-1β and IL-17. Glut1 is regulated by HIF-1α transcriptional activity and its elaboration is an important step in the metabolic switch from oxidative phosphorylation to glycolysis (*Chen et al., 2001*). Fluorodeoxyglucose positron emission tomography (FDG PET) scans are commonly used to identify metabolic activity in cancer and PET scans have been shown to be useful in active sarcoidosis (*Avril, 2004*; *Ben-Haim and Ell, 2009*). Increased Glut1 levels may explain the observed increased FDG uptake in PET/CT scans in active sarcoidosis (*Sobic-Saranovic et al., 2013*). Despite the importance of HIF signaling, the role of HIF-1α and HIF-2α in lung diseases has not been established and only a few studies addressed the role of HIFs in primary human immune cells. One previous study reported increased HIF-1α mRNA in lymphocytes of peripheral blood but a decreased mRNA level in HIF-1α BAL cells. In contrast to our study one prior study reported decreased HIF-1α mRNA and protein expression in sarcoidosis tissue biopsies (*Tzouvelekis et al., 2012*), although the same study reported increased expression of VEGF, which is directly regulated by HIF (*Tzouvelekis et al., 2012*). The discrepancy of the results may be due to stages of the disease or evaluation of heterogeneous cell populations.

Several pathways including the PI3 kinase, mTOR, MEK/ERK, GSK3β, and p38 pathways have been proposed to regulate LPS mediated HIF-1α expression and stabilization (*Palazon et al., 2014*; *Peyssonnaux et al., 2007*; *Talwar et al., 2017a*; *Talwar et al., 2019*). Previously, we have shown that sustained p38 activation directly controls expression of several cytokines in sarcoid AMs (*Rastogi et al., 2011*). The increased p38 phosphorylation in sarcoidosis was associated with lack of mitogen activated protein kinase phosphatase (MKP)-1 expression in sarcoidosis AMs and monocytes (*Rastogi et al., 2011*). Furthermore, p38 MAPK regulates IL-17 production by Th17 cells through regulation of various transcription factors (*Huang et al., 2015*; *Noubade et al., 2011*). Interestingly, our recent study showed that macrophages derived from MKP-1 deficient mice exhibited higher HIF-1α and IL-1β expression and higher ROS production in response to LPS; in addition, p38 inhibition decreased HIF-1α expression in MKP-1 deficient macrophages and modified cytokine production (*Talwar et al., 2017a*). In our current work, we found significantly higher HIF-1α expression in sarcoidosis AMs and PBMCs. This can be partly explained by a constitutively active p38 in macrophages of sarcoidosis subjects (*Rastogi et al., 2011*; *Talreja et al., 2016*). We observed that a p38 inhibitor (SB203580) partly decreased the expression of HIF-1α (data not shown) and cytokine levels in sarcoidosis.

Activation of TLR4 and TLR2 by a variety of pathogen-derived molecules as well as environmental toxins has been shown to induce and stabilize HIF-1α expression (*Frede et al., 2007*; *Liao et al., 2014*; *Palazon et al., 2014*). Abundance of HIF-1α in sarcoidosis also implies aberrant degradation by proteasomal or/and lysosomal pathways. Autophagy and the ubiquitin-proteasome system (UPS) are two major pathways involved in the degradation of proteins. It has been shown that there is a compensatory interaction between these two pathways and inhibition of one pathway leads to activation of the other (*Wang et al., 2013*). Our RNA sequencing data showed upregulation of lysosomal pathways, confirming previous findings by other investigators (*Talreja et al., 2016*; *Tomita et al., 1999*). LAMP2, along with LAMP1, comprise about 50% of lysosomal proteins. In sarcoidosis we observed upregulation of LAMP2 both at the gene and protein level. CHQ is an ancient drug that in addition to its anti-malaria activity has been used for autoimmune diseases, including sarcoidosis (*Morse et al., 1961*). Therefore, we determined the effect of CHQ on LAMP2 and HIF-α isoform expression. Surprisingly, we found that CHQ inhibits the increased levels of LAMP2, HIF-α isoforms, and cytokine production in sarcoidosis. We speculate that in sarcoidosis inhibition of

lysosomal function by CHQ leads to increased proteasome degradation of HIF-α isoforms leading to subsequent inhibition of IL-1β and IL-17 cytokines production.

Environmental factors, altered metabolism, and inflammation can be linked to epigenetic changes such as methylation and acetylation that may contribute to HIF-1α expression and stability in sarcoidosis (*Watson et al., 2010*). How HIF signaling in the absence of a hypoxic trigger regulates metabolic reprogramming and influences inflammation in chronic inflammatory diseases, especially respiratory diseases including sarcoidosis, has not been well illuminated. Our report identifies a role for HIF signaling in sarcoidosis granulomatous inflammation. The identification of the mechanisms underlying the aberrant regulation of HIF-1α and HIF-2α leading to persistent inflammation and Th1/Th17 pathology in sarcoidosis should open new avenues in rational drug discovery, not only for this disease but also for other inflammatory diseases.

## Materials and methods

| Reagent type (species) or resource | Designation | Source or reference | Identifiers | Additional information |
|---|---|---|---|---|
| Genetic reagent | Lipofectamine 2000 | Invitrogen | Cat. # 11668027 | |
| Biological sample | Human AMs | Bronchoalveolar lavage (BAL) cells | | Sarcoidosis Center, WSU, Detroit, USA |
| Biological sample | Human PBMCs and monocytes | Heparinized blood | | Sarcoidosis Center, WSU, Detroit, USA |
| Antibody | HIF-1α (rabbit polyclonal) | Bioss | RRID:AB_10857933, Cat. #bs0737 | WB (1:500) |
| Antibody | HIF-2α (rabbit polyclonal) | Bioss | RRID:AB_10857576, Cat. #bs1477 | WB (1:500) |
| Antibody | ARNT (rabbit monoclonal) | Cell Signaling | RRID:AB_2783880, Cat. # 5531 | WB (1:1000) |
| Antibody | P300 (rabbit polyclonal) | Santa Cruz Biotechnology | RRID:AB_2231120, Cat # sc-585 | WB (1:500) |
| Antibody | Glut1 (rabbit polyclonal) | Thermofisher Scientific | RRID:AB_2302087, Cat # PA1-46152 | WB (1:1000) |
| Antibody | LAMP2 (mouse monoclonal) | Santa Cruz Biotechnology | RRID:AB_626858, Cat # sc-18822 | WB (1:1000) |
| Antibody | pro-IL-1β (goat polyclonal) | R and D | Cat # AF-201-NA | WB (1:1000) |
| Antibody | β-actin (rabbit polyclonal) | Abcam | RRID:AB_2305186Cat # ab8227 | WB (1:1000) |
| Antibody | CD4-FITC (mouse monoclonal) | BD Biosciences | RRID:AB_400007, Cat # 340133 | |

*Continued on next page*

*Continued*

| Reagent type (species) or resource | Designation | Source or reference | Identifiers | Additional information |
|---|---|---|---|---|
| Antibody | CD25-PE (mouse monoclonal) | BD Biosciences | RRID:AB_400203, Cat # 341009 | |
| Antibody | CD14-PerCPCy5.5 (mouse monoclonal) | BD Biosciences | RRID:AB_2033939, Cat # 561116 | |
| Sequence-based reagent | HIF-1α siRNA, Sense | GGAACCUGAU GCUUUAACUtt | | Thermofisher-Scientific |
| Sequence-based reagent | HIF-1α siRNA, Anti-sense | AGUUAAAGCA UCAGGUUCCtt | | Thermofisher-Scientific |
| Commercial assay or kit | MTS assay kit ELISA kits | Promega R and D | | |
| Chemical compound, drug | Chloroquine | Invivo Gen | tlrl-chq | 100 mM |
| Chemical compound, drug | Echinomycin | Sigma | SML0477 | 10 nM |

## Chemicals

Chemicals were purchased from Sigma Chemical (St. Louis, MO) unless specified otherwise. LPS and chloroquine was purchased from InvivoGen (San Diego, CA). Antibodies against HIF-1α (# bs0737) and HIF-2α (#bs1477) were purchased from Bioss Inc (Woburn, MA), P300 (sc-585) was from Santa Cruz Biotechnology (Santa Cruz, CA), Glut1 (PA1-46152) from Thermofisher Scientific (Waltham, MA). The antibody for pro-IL-1β (# AF-201-NA) was purchased from R and D Systems (Minneapolis, MN), and β-actin (#ab8227) was purchased from Abcam (Cambridge, MA). Horseradish peroxidase–conjugated anti-mouse IgG (#7076S) and anti-rabbit IgG (#7074S) antibodies and antibody for ARNT (#5531) were purchased from Cell Signaling Technology (Beverly, MA). Horseradish peroxidase–conjugated anti-goat IgG (sc-2033) was purchased from Santa Cruz Biotechnology, The anti-human antibodies used for flow cytometry were CD4-FITC (#340133), CD25-PE (#341009), CD14-PerCPCy5.5 (#561116) and purified CD3 (#555337), purchased from BD Biosciences (San Jose, CA). The secondary antibody used for immunostaining Alexa 488 (#A11070) was purchased from Molecular Probes (Grand Island, NY). CellTiter 96 AQueous One Solution Cell Proliferation Assay was purchased from Promega (Madison, WI).

## Study Design

The Committee for Investigations Involving Human Subjects at Wayne State University approved the protocol for obtaining alveolar macrophages by bronchoalveolar lavage (BAL) and blood by phlebotomy from control subjects and patients with sarcoidosis. The IRB number for this study is 055208MP4E. All methods were performed in accordance with the relevant guidelines and regulations. Informed consent was obtained from all subjects enrolled for the study. Sarcoidosis diagnosis was based on the ATS/ERS/WASOG statement (*Hunninghake et al., 1999*). The criteria for enrollment in the diseased group were: (i) a compatible clinical/radiographic picture consistent with sarcoidosis, (ii) histologic demonstration of non-caseating granulomas on the tissue biopsy, and (iii) exclusion of other diseases capable of producing a similar histologic or clinical picture, such as fungus or mycobacteria. Subjects excluded were: (i) smokers, (ii) individuals receiving immune suppressive medication (defined as corticosteroid alone and/or in combination with immune modulatory medications), (iii) individuals with positive microbial culture in routine laboratory examinations or viral infection; or (iv) individuals with known hepatitis or HIV infections or any immune suppressive condition. The criteria for enrollment in the control group were: (i) absence of any chronic respiratory

diseases, (ii) lifetime nonsmoker, (iii) absence of HIV or hepatitis infection, and (iv) negative microbial culture. A total of 51 patients with sarcoidosis and 23 controls participated in this study. The medical records of all patients were reviewed, and data regarding demographics, radiographic stages, pulmonary function tests, and organ involvements were recorded.

### BAL and the preparation of alveolar macrophages (AMs)

BAL was collected during bronchoscopy after administration of local anesthesia and before tissue biopsies (*Rastogi et al., 2011*; *Talreja et al., 2016*). Briefly, a total of 150 to 200 mL of sterile saline solution was injected via fiberoptic bronchoscopy; the BAL fluid was retrieved, measured, and centrifuged. Cells recovered from the BAL fluid were filtered through a sterile gauze pad and washed three times with phosphate-buffered saline (PBS), resuspended in endotoxin-free RPMI 1640 medium (HyClone) supplemented with L-glutamine (Life Technologies), penicillin/streptomycin (Life Technologies), and 1% fetal calf serum (HyClone), and then counted. BAL cells were cultured on adherent plates for 1 hr at 37°C in air containing 5% $CO_2$. Non-adherent cells were removed by aspiration; adherent cells were washed three times and used as AMs. Viability of these populations was routinely about 97% and by morphologic criteria the adherent cells were in excess of 99% AMs (*Rastogi et al., 2011*; *Talreja et al., 2016*).

### Isolation of PBMCs and purification of monocytes

PBMCs were isolated from heparinized blood using Ficoll-Histopaque (Sigma, St. Louis, MO) density gradient separation followed by washing twice with PBS. Cell suspension was made in endotoxin-free RPMI 1640 medium (HyClone) supplemented with L-glutamine (Life Technologies), penicillin/streptomycin (Life Technologies), and 10% fetal calf serum (HyClone). Cells were cultured in 12-well plates for further experiments (*Rastogi et al., 2011*; *Talreja et al., 2016*). CD14$^+$ monocytes were purified from PBMCs by using the MACS monocyte isolation kit (Miltenyl Biotech, San Diego, CA) according to the manufacturer's instructions. The purity of enriched monocytes was evaluated by flow cytometry using PerCPCy5.5-conjugated CD14 antibody (#561116, BD Biosciences); the purity of monocytes was about 95%.

### Targeted down regulation of HIF-1α via siRNA

Isolated AMs or PBMCs were transiently transfected with non-specific silencer siRNA (NS siRNA, 200 pM) or targeted HIF-1α silencer siRNA (200 pM, Thermofisher-Scientific) in the presence of lipofectamine 2000 (Invitrogen). The sequence of siRNA used: sense (5′−3′) GGAACCUGAUGCUUUAACUtt and antisense AGUUAAAGCAUCAGGUUCCtt. After 24 hr of transfection, cells were activated with either LPS (100 ng/mL) or anti-CD3 (1 μg/mL). Viability of cells was assessed after siRNA treatment by MTS assay and 95% of cells were viable.

### Cell viability

Cell viability was assessed using MTS assay [CellTiter 96 AQueous One Solution Cell Proliferation Assay] (Promega, Madison, WI) following the manufacturer's instructions. Briefly, cells equivalent to $1 \times 10^4$/well were seeded in 96-well culture plate and incubated for 24–48 hr with different treatments. After incubation, 20 μl of CellTiter 96 AQueous One Solution Reagent was added per well for 2 hr and the absorbance was measured at 490 nm using a 96-well plate reader.

### Enzyme- Linked Immunosorbent Assay (ELISA)

The levels of IL-1β, IL-1Ra, IL-17, IL-10, IL-6, and IFN-γ in the conditioned medium were measured by ELISA according to the manufacturer's instructions (ELISA DuoKits; R and D Systems, Minneapolis, MN).

### Flow cytometry and cell surface immunostaining

PBMCs from subjects with sarcoidosis were isolated, cultured, and after appropriate treatment were stained for cell surface markers using fluorescent labelled antibodies for CD4-FITC (#340133, BD Biosciences), and CD25-PE (#341009, BD Biosciences). Intracellular staining of PBMCs was done for HIF-1α and HIF-2α. Briefly, PBMCs were first surface stained for CD14 using CD14-PerCPCy5.5 antibody and then fixed using 100 μl of 1% paraformaldehyde for 30 min and then permeabilized with

permeabilization buffer (0.5% saponin) for 15 min at room temperature. Cells were centrifuged and resuspended in 100 µl of permeabilization buffer and stained with HIF-1α (bs0737, Bioss Inc) or HIF-2α (bs1477, Bioss Inc) antibody for 30 min. Cells were washed and stained with the Alexa 488 secondary antibody (#A11070, Molecular Probes). After 30 min cells were washed twice, resuspended in staining buffer, and analyzed for CD14$^+$HIF-1α$^+$ and CD14+HIF-2α$^+$ monocytes by flow cytometry. PBMCs were not stained specifically to exclude DC contamination. Flow cytometry was performed on a BD LSR II SORP and data analysis was performed using FlowJo software (FlowJo, LLC, Ashland, OR) (*Talreja et al., 2016*). Samples were gated on cells using FSC/SSC and doublet discrimination was performed to identify singlets using SSC-W vs. SSC-A. The flowcytometry work was done at the Microscopy, Imaging and Cytometry Resources (MICR) Core at the Karmanos Cancer Institute, Wayne State University.

## Immunofluorescent staining

Intracellular expression of HIF-1α in sarcoidosis AMs was visualized by immunofluorescence staining. AMs (1 × 10$^5$) were allowed to adhere overnight on chamber slides. The cells were washed briefly with PBST and fixed with 3.7% paraformaldehyde. Cells were washed and permeabilized with 0.1% Triton X-100, blocked (10% FCS), and then incubated with anti-HIF-1α (bs0737,Bioss Inc) overnight at 4°C. The secondary antibody used was Alexa-fluor 488- conjugated goat anti-rabbit antibody. The next day cells were washed three times with PBS for 5 min, the slide was mounted with a drop of ProLong Gold antifade reagent with DAPI (Invitrogen). The slide was examined using an Axiovert 40 CFL fluorescence microscope (Carl Zeiss MicroImaging, Inc).

## Protein extraction and immunoblotting

Total cellular proteins were extracted by addition of RIPA buffer containing a protease inhibitor cocktail and antiphosphatase I and II (Sigma Chemicals). Protein concentration was measured with the BCA assay (Thermo Scientific, CA). Equal amounts of proteins (10–25 µg) were mixed 1/1 (v/v) with 2x sample buffer (20% glycerol, 4% sodium dodecyl sulfate, 10% 2-βME, 0.05% bromophenol blue, and 1.25 M Tris-HCl, pH 6.8), and fractionated on a 10% sodium dodecyl sulfate–polyacrylamide gel. Proteins were transferred onto a polyvinylidene difluoride membrane (Bio-Rad) for 60 min at 18 V using a SemiDry Transfer Cell (Bio-Rad). The polyvinylidene difluoride membrane was blocked with 5% nonfat dry milk in TBST (Tris-buffered saline with 0.1% Tween 20) for 1 hr, washed, and incubated with primary Abs (1/1000) overnight at 4°C. The blots were washed with TBST and then incubated for 1 hr with horseradish peroxidase–conjugated secondary anti-IgG Ab using a dilution of 1/10,000 in 5% nonfat dry milk in TBST. Membranes were washed four times in TBST. Immuno-reactive bands were visualized with a chemiluminescent reagent (Amersham GE, PA). Images were captured on Hyblot CL film (Denville; Scientific, Inc; Metuchen, NJ) using a JPI automatic X-ray film processor model JP-33. Optical density analysis of signals was performed using Image J software (*Rastogi et al., 2011*; *Talreja et al., 2016*).

## Immunohistochemistry

Tissue sections from the sarcoidosis transbronchial lung biopsy samples were selected for immunostaining after review of the glass slides that had been previously prepared using the routine hematoxylin-eosin protocol on paraffin-embedded sections. Additional fixed slides were cut, subjected to peroxide block protocol, pretreated, and then incubated first with primary antibody (anti-HIF-1α, bs0737, Bioss Inc ) and then with a secondary conjugated polymer; each incubation step was done for 30 min at room temperature. Negative staining was done by using an isotype control antibody. After another incubation step with the chromogen (5 min at room temperature), the sections were counterstained with hematoxylin and dehydrated with ethanol and xylene prior to mounting. Images were analyzed by microscopy (BX40, Olympus).

## Statistical Analyses

A Student *t-test* or one-way analysis of variance and *post hoc* repeated measure comparisons (least significant difference) were performed to identify differences between groups. ELISA results were expressed as mean ± SEM. For all analyses, two-tailed *p* values of less than 0.05 were considered to be significant.

## Acknowledgements

This work was supported by grants from NIH (R01HL113508) (LS) and the American Lung Association (LS) and as well as the Department of Medicine and the Center for Molecular Medicine and Genetics, Wayne State University School of Medicine (LS). The Microscopy, Imaging and Cytometry Resources Core is supported, in part, by NIH Center grant P30 CA022453 to the Karmanos Cancer Institute at Wayne State University and the Perinatology Research Branch of the National Institutes of Child Health and Development at Wayne State University. LIG is supported by the Office of the Assistant Secretary of Defense for Health Affairs through the Peer Reviewed Medical Research Program under Award W81XWH-16-1-0516 and the Henry L Brasza endowment at Wayne State University. The views expressed in this article are those of the authors and do not necessarily reflect the position or policy of the US Department of Defense or the United States government.

## Additional information

### Funding

| Funder | Grant reference number | Author |
|---|---|---|
| U.S. Department of Defense | W81XWH-16-1-0516 | Lawrence I Grossman |
| Wayne State University | Henry L Brasza endowment | Lawrence I Grossman |
| National Heart, Lung, and Blood Institute | R01HL113508 | Lobelia Samavati |
| American Lung Association | | Lobelia Samavati |
| School of Medicine, Wayne State University | | Lobelia Samavati |

The funders had no role in study design, data collection and interpretation, or the decision to submit the work for publication.

### Author contributions

Jaya Talreja, Data curation, Formal analysis, Visualization, Methodology, Writing—original draft, Writing—review and editing; Harvinder Talwar, Data curation, Visualization, Methodology, Writing—review; Christian Bauerfeld, Formal analysis, Visualization, Methodology, Writing—review and editing; Lawrence I Grossman, Visualization, Writing—review and editing; Kezhong Zhang, Writing—review and editing; Paul Tranchida, Methodology; Lobelia Samavati, Conceptualization, Resources, Formal analysis, Supervision, Funding acquisition, Validation, Visualization, Project administration, Writing—review and editing

### Author ORCIDs

Jaya Talreja (iD) https://orcid.org/0000-0001-6557-6500
Lobelia Samavati (iD) http://orcid.org/0000-0002-3327-2585

### Ethics

Human subjects: The Committee for Investigations Involving Human Subjects at Wayne State University approved the protocol for obtaining alveolar macrophages by bronchoalveolar lavage (BAL) and blood by phlebotomy from control subjects and patients with sarcoidosis. The IRB number for this study is 055208MP4E. Informed consent was obtained from all subjects enrolled for the study.

### Decision letter and Author response

Decision letter https://doi.org/10.7554/eLife.44519.025
Author response https://doi.org/10.7554/eLife.44519.026

## Additional files

### Supplementary files
• Transparent reporting form
DOI: https://doi.org/10.7554/eLife.44519.020

### Data availability
All data generated or analysed during this study are included in the manuscript.

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
