## [Decision Letter]

Thank you for submitting your article "HIF-1α Regulates IL-1β and IL-17 in Sarcoidosis" for consideration by *eLife*. Your article has been reviewed by three peer reviewers, including Jos WM van der Meer as the Reviewing Editor and Reviewer #1, and the evaluation has been overseen by Tadatsugu Taniguchi as the Senior Editor. The following individual involved in review of your submission has agreed to reveal their identity: Siroon Bekkering (Reviewer #2).

The reviewers have discussed the reviews with one another and the Reviewing Editor has drafted this decision to help you prepare a revised submission.

Summary:

This is a well-executed study showing a key role for unregulated HIF-1α in the pro-inflammatory cytokine response in sarcoidosis. Mechanistic studies in sarcoidosis are definitely needed, since the understanding of the pathogenesis is still rather limited. The clinical material used, the experiments both in alveolar macrophages and in peripheral monocytes are adequate and convincing, also because of the inhibition studies performed.

Essential revisions:

1) The authors assert that, to the best of their knowledge, this is the first report to interrogate the role of HIF in sarcoidosis. Whilst this is the most comprehensive, and likely a more robust study to look at the function of HIF in the disease, there are two other reports detailing HIF expression in sarcoidosis (Tzouvelekis et al., 2012 and Piotrowski et al., 2015). One of these studies conflicts with the finding of HIF granuloma expression presented here in this report, therefore it would be of value to discuss these within this paper and provide possible reasons for the differences or similarities in findings.

2) Regarding flow cytometry, the authors show HIF-1α expression in CD14^+^ cells. They should FSC and SSC gating and acknowledge DC contamination, if that has not been dealt with.

3) The conclusion drawn from flow cytometry data for PBMCs makes reference to increased CD14^+^HIF-1α^+^monocytes in the patient samples, stating that this is 5-9% in controls and 20-35% in sarcoidosis patient samples. This is misleading, as monocytes make up a higher percentage of PBMCs in sarcoidosis. It is suggested that this be reworded to represent HIF^+^ percentage of monocytes, and that quantitative figures provided for all donors along with degree of expression (MFI or equivalent).

4) For Figure 3, a negative antibody control staining is necessary to support the conclusion, as macrophages and giant cells display high non-specific staining. This is important for validation in light of point 1.

5) The authors show single cell examples of confocal microscopic studies. The question is are all cells showing this upregulated pattern? A remark on this would be helpful.

6) The authors say that they have observed similar results on liver and skin samples (subsection “Confocal microscopy of sarcoidosis AMs and immunohistochemistry of sarcoidosis tissues confirmed increased HIF-1α expression and its nuclear accumulation”). This is a bit too loose. How many samples? What does 'similar' exactly mean?

7) In Figure 4, why were the cells not stimulated by LPS and why was only baseline expression measured? In Figure 5, the expression is measured after stimulation and it might be interesting to look at secreted IL-1β after stimulation in Figure 4 as well. Furthermore, in Figure 5C, the expression of pro-IL-1β (unstimulated pro-IL-1β) should be similar to the expression in Figure 4 (as it is the same measurement) but suddenly looks much lower. Why is this?

8) Was the viability of the cells checked after siRNA treatment? Primary cells usually don't like siRNA treatment and die. Please add proof of viability.

9) In Figure 6B, the expression of IL-10 looks decreased. There is no statistical significance (as is hard to obtain with n=4 generally), but the same trend is seen in Figure 6A. What would happen if one would increase it to n=6, would that become statistically significant? Will that change the conclusions?

10) In discussing the specificity of HIF for IL-17 and IL-1β, the data shows an impact on IL-6 also. Additionally, in discussions, the lack of significant impact on IFN-γ is noted but not considered of importance. IL-6 and IFN-γ are notably important cytokines in sarcoidosis, both being overexpressed in granulomatous tissue, so addressing these findings would be useful for wider context.

11) In the text explaining Figure 7A, it is mentioned that patients’ PBMCs have a higher expression of CD4^+^CD25^+^ cells. But higher than what? There are no control PBMCs included in this analysis, is that correct? Why not? What is the baseline expression in control PBMCs?

12) The findings of sustained enhanced (spontaneous) IL-1 production and upregulated HIF-1α are reminiscent of the trained immunity state of mononuclear phagocytes, in which the metabolic shift (Warburg effect) and epigenetic reprogramming are tightly connected (see Cheng et al., 2014). This should be discussed.

13) When IL-1β production is enhanced, the net biological effect is often dependent on the production level of IL-1Ra. If the investigators still have supernatants, it would be of interest to see whether IL-1Ra is also unregulated.

14) A main criticism is the Discussion. This section could be much better structured and written, as the reviewers lost track of the authors' findings in their reasoning. Please start with a few sentences telling the key findings of this work, and put that into the perspective of what is known about this in sarcoidosis and in general. Thereafter, please go into the minor findings, to end with future perspectives (and speculate about the clinical implications of these very interesting findings).

15) In the Discussion, two interesting results of inhibitors used are mentioned with 'data not shown'. Why are these data not included in the manuscript? This would add some interesting pathway information to the story. It is preferable not to mention unpublished data, if at all possible.

References:

Piotrowski WJ, Kiszałkiewicz J, Pastuszak-Lewandoska D, Górski P, Antczak A, Migdalska-Sęk M, Górski W, Czarnecka KH, Domańska D, Nawrot E, Brzeziańska-Lasota E. Expression of HIF-1A/VEGF/ING-4 Axis in Pulmonary Sarcoidosis. Adv Exp Med Biol. 2015;866:61-9. doi: 10.1007/5584_2015_144.

[Editors' note: further revisions were requested prior to acceptance, as described below.]

Thank you for resubmitting your work entitled "HIF-1α Regulates IL-1β and IL-17 in Sarcoidosis" for further consideration at *eLife*. Your revised article has been favorably evaluated by Tadatsugu Taniguchi as the Senior Editor, and by Jos WM van der Meer as the Reviewing Editor.

The manuscript has been improved but there are some remaining issues that need to be addressed before acceptance, as outlined below:

1) The rebuttal to our major criticism 1 deserves a little more discussion. The rebuttal is quite OK but too little of it reached the revised manuscript. So please also incorporate the discrepancies with the Piotrowski paper.

2) The figure presented in the rebuttal to our criticism #6 merits to be added as a supplementary figure with a short(er) description of the case in the legend.

3) The rebuttal under criticism #9 should also be reflected in the revision.

4) Give the full reference for the Avril paper.

---

## [Author Response]

Essential revisions:1) The authors assert that, to the best of their knowledge, this is the first report to interrogate the role of HIF in sarcoidosis. Whilst this is the most comprehensive, and likely a more robust study to look at the function of HIF in the disease, there are two other reports detailing HIF expression in sarcoidosis (Tzouvelekis et al., 2012 and Piotrowski et al., 2015). One of these studies conflicts with the finding of HIF granuloma expression presented here in this report, therefore it would be of value to discuss these within this paper and provide possible reasons for the differences or similarities in findings.

We were aware of these two publications. Both studies evaluated predominantly HIF-1α mRNA levels using PCR. Tzouvelekis et al. reported decreased expression of HIF-1α mRNA and protein in tissue biopsies. The tissue presented in that manuscript appears to show advanced fibrotic changes with complete destruction of lung structures. Furthermore, they show upregulation of VEGF expression. Increased VEGF expression indicates increased HIF-1α activity as VEGF is highly regulated by HIF-1α. The discrepancy between our results and their findings could be due to sampling of biopsy at different stages of the disease or treatment effect. Note that in our study, most of the patients were recruited before starting any drug treatment. Furthermore, similar to their results our RNA-Seq data didn’t show any differential expression of HIF-1α between healthy controls and sarcoid patients. We mentioned that pathway analysis indicated upregulation of metabolic pathways that are regulated by HIF pathways. The study done by Piotrowski et al. was concluded that the upregulation of HIF-1α and VEGF in PBMCs of sarcoid patients was associated with poor lung function. We were unable to access the complete paper of Piotrowski. Reading the Abstract, it is not clear to us what cell type they analyzed (blood lymphocytes versus BAL?). Hence, we are unsure how those results compare to our findings.

2) Regarding flow cytometry, the authors show HIF-1α expression in CD14^+^ cells. They should FSC and SSC gating and acknowledge DC contamination, if that has not been dealt with.

As suggested, we have shown the FSC and SSC gating in Figure 2J. We acknowledge that we did not do any staining to exclude DCs. For flowcytometry, PBMCs were double stained with CD14 and HIF-1α antibodies. The upper right quadrant shows CD14^+^HIF-1α^+^ cells that are CD14^+^monocytes expressing HIF-1α. We added a sentence indicating that we did not specifically assess for DCs.

3) The conclusion drawn from flow cytometry data for PBMCs makes reference to increased CD14^+^HIF-1α^+^monocytes in the patient samples, stating that this is 5-9% in controls and 20-35% in sarcoidosis patient samples. This is misleading, as monocytes make up a higher percentage of PBMCs in sarcoidosis. It is suggested that this be reworded to represent HIF^+^ percentage of monocytes, and that quantitative figures provided for all donors along with degree of expression (MFI or equivalent).

We agree with the comment. We have reworded in the Results and legend sections to accommodate your suggestion, however we also think it important to mention double positivity of CD14 and HIF-1α.

The percentage of CD14^+^HIF-1α^+^monocytes obtained from the donors used for the study are shown in Author response image 1.

4) For Figure 3, a negative antibody control staining is necessary to support the conclusion, as macrophages and giant cells display high non-specific staining. This is important for validation in light of point 1.

It is our opinion that the best control is the unaffected appearing lung tissue of the same subject. However, we added the negative (isotype control) staining as suggested in Figure 3K.

5) The authors show single cell examples of confocal microscopic studies. The question is are all cells showing this upregulated pattern? A remark on this would be helpful.

Thank you for your comments. The purpose of performing confocal microscopy of single cells was to identify whether HIF-1α is present in the nuclei, suggesting transcriptional activity of HIF-1α. Furthermore, due to the fact that we worked with precious human samples, and we could not perform fractionation of nuclear proteins (due to limited number of cells), we performed confocal microscopy instead. Based on the reviewer’s suggestion we performed regular immunofluorescent microscopy and quantitated the percentage of cells showing HIF-1α expression. It shows that about 60-90% of AMs show HIF-1α expression. We added a representative section of DAPI and HIF-1α in Figure 3.

6) The authors say that they have observed similar results on liver and skin samples (subsection “Confocal microscopy of sarcoidosis AMs and immunohistochemistry of sarcoidosis tissues confirmed increased HIF-1α expression and its nuclear accumulation”). This is a bit too loose. How many samples? What does 'similar' exactly mean?

We assessed HIF-1α expression in two skin and two liver biopsies. In one liver biopsy, we noticed high HIF-1 a staining in granulomatous tissue and surrounding hepatocytes. This patient is a young woman (28 year old), who at 21 years of age developed sarcoidosis. She has severe multiorgan involvement, including lungs, eyes and heart’ as well as liver. Due to sarcoidosis uveitis she is now legally blind. She had very abnormal liver function tests. The purpose of performing a liver biopsy was to assess for her abnormal liver function tests. H&E staining showed severe granulomatous inflammation. There were extensive increased HIF-1α expression throughout the liver. While negative staining did not show non-specific staining. Therefore, we believe that increased HIF-1α staining in the hepatocytes may represent a true increased of HIF-1α associated with the severity of disease in this patient. We have now added Figure 3—figure supplement 1 in the final version to show there are multiple granulomas in the liver with increased mononuclear cell infiltrates. Yet, no significant fibrosis is seen in the H&E staining.

7) In Figure 4, why were the cells not stimulated by LPS and why was only baseline expression measured? In Figure 5, the expression is measured after stimulation and it might be interesting to look at secreted IL-1β after stimulation in Figure 4 as well.

In Figure 4, we are showing the comparison between the baseline levels of pro- IL-1β between sarcoid patients and healthy controls. Therefore, we selected the unstimulated samples of patients and controls. As suggested by the reviewers we have added the data for secreted IL-1β with LPS stimulation in revised Figure 4F and 4G.

Furthermore, in Figure 5C, the expression of pro-IL-1β (unstimulated pro-IL-1β) should be similar to the expression in Figure 4 (as it is the same measurement) but suddenly looks much lower. Why is this?

Figure 5C and Figure 4, Western blots are generated from different patient samples. As such, these are different biological replicates and not from the same subject. In Figure 5C, we used the lower exposure blot to show the changes in the pro-IL-1β levels after LPS stimulation and HIF-1α siRNA treatment. Author response image 2 is the Western blot of the same samples with higher exposure.

**Author response image 2. respfig2:** 

8) Was the viability of the cells checked after siRNA treatment? Primary cells usually don't like siRNA treatment and die. Please add proof of viability.

We usually assess cell viability in multiple ways after each experiment: (1) inspection by light microscopy, (2) trypan blue staining, and (3) MTT or MTS assay. The viability of the cells after siRNA treatment was determined by MTS assay (Promega). Both the AMs and PBMCs were viable after the siRNA treatment as shown in Author response image 3.

**Author response image 3. respfig3:** 

9) In Figure 6B, the expression of IL-10 looks decreased. There is no statistical significance (as is hard to obtain with n=4 generally), but the same trend is seen in Figure 6A. What would happen if one would increase it to n=6, would that become statistically significant? Will that change the conclusions?

We agree that in Figure 6B, IL-10 trends to be lower after treatment of PBMCs with HIF-1α siRNA, although not statistically significant. Similarly, in Figure 5 the treatment of AMs with HIF-1α siRNA didn’t show a statistically significant decrease in IL-10. We agree that both Interferon and IL-10 show a trend to lower levels, which can be due to paracrine IL-1β function.

10) In discussing the specificity of HIF for IL-17 and IL-1β, the data shows an impact on IL-6 also. Additionally, in discussions, the lack of significant impact on IFN-γ is noted but not considered of importance. IL-6 and IFN-γ are notably important cytokines in sarcoidosis, both being overexpressed in granulomatous tissue, so addressing these findings would be useful for wider context.

Thank you for the comment. Our current results show the co-regulation of IL-1β and IL-6 confirming our previous published results in sarcoidosis (Talreja et al., 2016). In this manuscript, we chose to be focused on IL-1β and IL-17. Based on reviewers’ suggestion we added a sentence to this fact.

11) In the text explaining Figure 7A, it is mentioned that patients’ PBMCs have a higher expression of CD4^+^CD25^+^ cells. But higher than what? There are no control PBMCs included in this analysis, is that correct? Why not? What is the baseline expression in control PBMCs?

We agree with reviewers that in the text we should mention the baseline expression of CD4^+^CD25^+^ cells in control and sarcoid PBMCs. In fact, we have analyzed 23 PBMCs samples from sarcoid patients and 7 samples from healthy controls. The range of baseline expression of CD4^+^CD25^+^ cells in control PBMCs is about 1-5% whereas in sarcoid patients it is about 2-25%.

After reviewing our data, we would like to replace the previous figure with results of a different patient, which is more representative of the study group. Therefore, we added a new Figure 7A-D, as this is more representative of sarcoidosis subjects.

In response to your question, we would like to provide a box plot of CD4^+^CD25^+^ cells in control and sarcoid PBMCs obtained using flowcytometry. In the revised manuscript, we have provided these data in the Results section.

**Author response image 4. respfig4:** 

12) The findings of sustained enhanced (spontaneous) IL-1 production and upregulated HIF-1α are reminiscent of the trained immunity state of mononuclear phagocytes, in which the metabolic shift (Warburg effect) and epigenetic reprogramming are tightly connected (see Cheng et al., 2014). This should be discussed.

We added that in the Discussion.

13) When IL-1β production is enhanced, the net biological effect is often dependent on the production level of IL-1Ra. If the investigators still have supernatants, it would be of interest to see whether IL-1Ra is also unregulated.

We agree with the reviewers that it is interesting to see whether IL-1Ra is also unregulated when IL-1β production is enhanced. As suggested, we have measured levels of IL-1Ra via ELISA in the supernatants of cultured AMs and PBMCs from patients and controls samples. We included IL-1Ra data in revised Figure 4H and 4I. Although, several previous studies have already reported increased IL-1Ra in sarcoidosis.

14) A main criticism is the Discussion. This section could be much better structured and written, as the reviewers lost track of the authors' findings in their reasoning. Please start with a few sentences telling the key findings of this work, and put that into the perspective of what is known about this in sarcoidosis and in general. Thereafter, please go into the minor findings, to end with future perspectives (and speculate about the clinical implications of these very interesting findings).

We are thankful to reviewers for their valuable suggestions. We have revised the Discussion accordingly.

15) In the Discussion, two interesting results of inhibitors used are mentioned with 'data not shown'. Why are these data not included in the manuscript? This would add some interesting pathway information to the story. It is preferable not to mention unpublished data, if at all possible.

We eliminated the statement regarding dual IRAK1/4 and RIP2 inhibitor. We previously reported the importance of p38 activation in sarcoidosis. Similarly, we and other groups have shown that p38 inhibitor decreases the level of HIF-1α. We felt we should connect this result with our previous finding, but we did not see any novelty to provide the data.

[Editors' note: further revisions were requested prior to acceptance, as described below.]The manuscript has been improved but there are some remaining issues that need to be addressed before acceptance, as outlined below:1) The rebuttal to our major criticism 1 deserves a little more discussion. The rebuttal is quite OK but too little of it reached the revised manuscript. So please also incorporate the discrepancies with the Piotrowski paper.

We expanded the discussion about these two manuscripts.

2) The figure presented in the rebuttal to our criticism #6 merits to be added as a supplementary figure with a short(er) description of the case in the legend.

We added the liver biopsy result in Figure 3—figure supplement 1, with description.

3) The rebuttal under criticism #9 should also be reflected in the revision.

This is added in the Discussion.

4) Give the full reference for the Avril paper.

It is now provided.